# Modeling a hot, dry future: Substantial range reductions in suitable environment projected under climate change for a semiarid riparian predator guild

**Brian R. Blais**◯*, **John L. Koprowski**¤

School of Natural Resources and the Environment, University of Arizona, Tucson, Arizona, United States of America

¤ Current address: Haub School of Environment and Natural Resources, University of Wyoming, Laramie, Wyoming, United States of America

* bblais@arizona.edu

**Data Availability Statement:** Data are provided within the paper, Supporting Information files, and repository [https://figshare.com/s/

## Abstract

An understanding of species-environmental relationships is invaluable for effective conservation and management under anthropogenic climate change, especially for biodiversity hotspots such as riparian habitats. Species distribution models (SDMs) assess present species-environmental relationships which can project potential suitable environments through space and time. An understanding of environmental factors associated with distributions can guide conservation management strategies under a changing climate. We generated 260 ensemble SDMs for five species of *Thamnophis* gartersnakes (n = 347)—an important riparian predator guild—in a semiarid and biogeographically diverse region under impact from climate change (Arizona, United States). We modeled present species-environmental relationships and projected changes to suitable environment under 12 future climate scenarios per species, including the most and least optimistic greenhouse gas emission pathways, through 2100. We found that *Thamnophis* likely advanced northward since the turn of the 20th century and overwinter temperature and seasonal precipitation best explained present distributions. Future ranges of suitable environment for *Thamnophis* are projected to decrease by ca. -37.1% on average. We found that species already threatened with extinction or those with warm trailing-edge populations likely face the greatest loss of suitable environment, including near or complete loss of suitable environment. Future climate scenarios suggest an upward advance of suitable environment around montane areas for some low to mid-elevation species, which may create pressures to ascend. The most suitable environmental areas projected here can be used to identify potential safe zones to prioritize conservation refuges, including applicable critical habitat designations. By bounding the climate pathway extremes to, we reduce SDM uncertainties and provide valuable information to help conservation practitioners mitigate climate-induced threats to species. Implementing informed conservation actions is paramount for sustaining biodiversity in important aridland riparian systems as the climate warms and dries.

4833e20471b817e68c7f]. Occurrence data species with designations under the Endangered Species Act cannot be shared publicly due to the sensitive nature of those data; inquiries for those data can be directed to Arizona Game and Fish Department, Heritage Data Management System (HDMS) Program Manager at 623 236-7618 or HDMS@azgfd.gov. Additional inquiries can be directed to the corresponding author.

**Funding:** The author(s) received no specific funding for this work.

**Competing interests:** The authors have declared that no competing interests exist.

## Introduction

An understanding of species-environmental relationships can help to predict potential outcomes under shifting future scenarios [1–3], which is invaluable for conserving biodiversity during anthropogenic climate change [4,5]. Globally, climatic increases in aridity and temperature will likely place added pressure on water availability and demand [6]. Aridland ecosystems, such as the southwestern United States, are predicted to become warmer and drier [7–10]. Such changes can affect ecosystems through increased frequencies of major disturbances and extreme climatic events [11], especially in biodiverse riparian zones [12–14]. Slower, gradual changes to habitats are also occurring [15–17]. Environmental changes can adversely affect biota [18–25], and species must either adapt to changes in place, disperse to areas with favorable environment, or go extinct [4,26,27]. How and where species respond to challenges are of key importance for the maintenance of biodiversity [28–33].

Analyzing and predicting species distributions based on environmental gradients is required to decipher patterns in ecological niches [34–36]. Species distribution models (SDMs; *syn*. climate envelope models, ecological/environmental niche models, habitat suitability models; [37]) assess the concurrent relationships between species occurrence and environmental conditions (i.e., realized niche), which can then be modeled to infer and project (i.e., forecast) potential suitable environments through space and time [3,38,39]. Numerous tools, steps, and pathways exist to facilitate SDMs for a given set of parameters and objectives [40–42], and novel technological advances enable conservation practitioners to develop customized applications that can be evaluated across multiple scenarios [43–47]. Such resources are essential to guide effective conservation planning in light of climate change and extinction risks faced by vulnerable species [41,48,49].

Gartersnakes (*Thamnophis* spp.) are distributed throughout the Americas, usually have life histories connected to surface water, and are often considered riparian-dependent [50]. *Thamnophis* comprise an intermediate predatory guild of generalists and specialists and are important in energy transfer between aquatic and terrestrial habitats that help uphold ecosystem integrity [51,52]. Some *Thamnophis* are also sensitive to disturbances and changes to habitat and environment [53–57]. Given their trophic position, environmental sensitivity, and occupancy of important areas of biodiversity (e.g., riparian zones), *Thamnophis* make ideal model species to assess bioclimatic influences on contemporary distributions and project how climate change patterns may impact future distributions. Less is known about the environmental influences on *Thamnophis* in aridland regions.

Herein, we were interested in how contemporary and future climatic trends may impact environmentally sensitive aridland riparian species in a warming and drying climate (Arizona, United States). We used *Thamnophis* as a model system to address the following objectives: 1) examine spatial distributions and patterns in recent historical and contemporary occurrences; 2) assess explanatory relationships of bioclimatic variables on present distributions via ensemble species distribution models; and 3) use forecasting SDMs to project spatiotemporal changes of suitable environmental range under multiple future climate scenarios. Given the contemporary trends and likelihood of increased heat and drought in the southwestern U.S. region, we hypothesized reductions in range of suitable environmental for *Thamnophis* across time. We also expected to see reflections of upward advancement—in both elevation and latitude—as observed with numerous species impacted by climate change [58–61]. Coupled with ongoing challenges faced by *Thamnophis* in aridland climates [53,54,56,62], contraction of suitable environment will likely exacerbate risks to population demographics [63]. Our SDM assessments can help guide multi-species conservation management decisions, especially for taxa already under threat of extinction. More broadly, *Thamnophis* are important indicator

species and their relationships and responses to environmental change can serve as an early warning system for how other aridland riparian-linked species with similar life histories may respond to climate change.

## Materials and methods

### Study area and system

In the southwestern United States, the semiarid state of Arizona is biogeographically diverse and experiences a wide range of seasonally variable climatic gradients, including seasonally bimodal precipitation patterns with periods of drought fluctuated between rainy winters and summers [64–66]. Increases in aridity and temperature are predicted for the region, with less frequent but more intense precipitation trends likely to further strain natural resources [7,10,67–69]. These characteristics make Arizona an ideal study area to model present and future impacts of environmental change on representative taxa. Assessment at the state extent may also facilitate collaboration and extrapolation of conservation management applications among both localized and broader regional levels.

Five species of *Thamnophis* occur in Arizona; three are considered common (*T. cyrtopsis* = black-necked gartersnake; *T. elegans* = terrestrial gartersnake; and *T. marcianus* = checkered gartersnake) and two rare species that have threatened status at state and federal levels (*T. eques* = Mexican gartersnake; *T. rufipunctatus* = narrow-headed gartersnake; [56,62,63]). Their primarily allopatric distributions range from low desert drainage networks to subalpine conifer forest biomes, though some interspecific syntopy occurs [50,56,70]. Comprehensive life and natural history summaries of *Thamnophis* in Arizona is covered elsewhere [56].

### Occurrence data and historical spatial trends

We obtained georeferenced occurrence data for *Thamnophis* in Arizona from museum records (e.g., Vertnet), the global biodiversity information facility (GBIF, [71]), iNaturalist, field projects, and technical reports; records spanned from the 1800s through 2021. We manually inspected records with geocoordinate uncertainty ≥0.5 km—a conservative proxy of home range size for the focal taxa (ca. ≤10 ha [50,56,70])—and retained records where metadata descriptions were sufficiently detailed to confirm localities. Otherwise, we omitted imprecise records (>0.5 km) with vague descriptions, those with obscured/private locality masks, and source cross-duplicates. We also omitted highly suspect localities (see [56]).

To estimate *Thamnophis* range shifts in recent history, we first partitioned the occurrences dataset into three time periods. Records prior to 1940 reflect the pioneering days of herpetological exploration [72], 1940–1979 characterize increased efforts to supply herpetological collections [73], and 1980–2021 represent contemporary ("present") records (Table 1). We mapped occurrences and used the Directional Distribution tool in ArcMap v.10.8 (ESRI, Redlands, CA) to estimate the central tendency, dispersion, and rotational trends per historical period, then calculated distance and directional change among mean centers per period. We also estimated the relative percent change in standard deviational ellipse area, i.e., expansion or contraction in occurrence sampling patterns, by dividing the standard distance ellipse area (X × Y) among time periods. We combined all *Thamnophis* species for historical tests to increase sample size and infer broad spatial change patterns reflected by congenerics in the same region.

### Environmental data and global climate models

To assess present-time environmental influences on *Thamnophis* distributions, we obtained elevation and (19) bioclimatic predictor variables at 0.5′ (~1 km$^2$) spatial resolution from the

**Table 1. Georeferenced occurrences for the five *Thamnophis* gartersnake species recorded in Arizona through 31 December 2021.** Data sources include museum and biodiversity repositories, iNaturalist, agency reports, and field projects. Data was manually inspected for duplicates, erroneous records, and location accuracy (≤0.5 km).

| Gartersnake Species in Arizona | Pre-1940 | 1940–1979 | 1980–2019[‡] |
|---|---|---|---|
| *Thamnophis cyrtopsis*–black-necked gartersnakes | 17 | 54 | 180 (127) |
| *Thamnophis elegans*–terrestrial gartersnake | 13 | 37 | 170 (114) |
| *Thamnophis eques*–Mexican gartersnake[T] | 5 | 11 | 26 (20) |
| *Thamnophis marcianus*–checkered gartersnake | 3 | 37 | 83 (58) |
| *Thamnophis rufipunctatus*–narrow-headed gartersnake[T] | 2 | 21 | 476 (28) |

[‡]Numbers in parentheses indicate occurrence thinning to 1 per km²; [T] = "threatened" status at Arizona and Federal levels.

WorldClim Version 2 dataset [74]. We focused on bioclimatic variables and elevation—which is climatically linked [64]—for their explanatory power in SDM predictions and likelihood to drive changes to species distributions [38,75]. We used ArcMap to crop predictors to the spatial extent of the Arizona state boundary.

For future projections, we obtained the same bioclimatic variables from WorldClim v.2 that have been projected and downscaled from the Coupled Model Intercomparison Project 6th assessment (CMIP6) data [6]. We selected bioclimatic data from the following three global climate models (GCMs): HadGEM3-GC31-LL ("HAD"), MPI-ESM1-2-LR ("MPI"), and MRI-ESM2-0 ("MRI"); these GCMs have performed well for SDMs in North America, including representation of the North American Monsoon that drives summer climate in our study region [75–79]. For each GCM, we assessed two shared socio-economic pathways (SSPs): low reflected various emission projections (SSP126, i.e., progressive action towards sustainable development to limit ca. +1.8°C by 2100) and high emission projections (SSP585, ca. +4.4°C by 2100, i.e., *status quo* trajectory of continued societal and energy practices; [6]). These SSPs reflected the most optimistic and pessimistic pathways, respectively [80], and have been used in SDMs of southwestern herpetofauna [75,79]. For each SSP per GCM, we assessed two future time periods: near-future 2041–2060 (hereafter "2050", i.e., median year) and distant future 2081–2100 (hereafter "2090"). By assessing twelve various combinations of GCMs × SSPs × time periods (hereafter "scenarios"), including best-case/worst-case pathways, we better represent uncertainties in projections that should outweigh loss of precision generated from a single model scenario [79,81].

## Species distribution modeling and evaluation

We used the R package biomod2 version 4.2–2 [43,82] to generate SDMs separately per species. We built "present" scenario SDMs with only 1980–2021 records to avoid erroneously assessing since-altered habitat or local extirpations (see [56]). We used the spThin package [83] to delimit occurrences to 1/km² separately for each species (Table 1; [84]); this scale corresponded to the environmental layers. We created 1,000 randomly distributed pseudo-absences, i.e., background points representing available environment [85,86], per modeling run. Thickening the number of background points greater than presence points has performed well in SDMs [87], especially when taxa have limited sample size [88].

We ran preliminary modeling procedures (see below) for each species separately with all bioclimatic predictors except those comprising precipitation-temperature interactions which are problematic with discontinuities [89], especially in regions with bimodal precipitation patterns like Arizona. We further reduced the number of predictors in three ways based on

collinearity (omittance |r| > 0.7; [90]), ranked variable importance scores from biomod2 to break ties among correlated pairs [91], and *a priori* knowledge from the literature and expertise of the focal system [92]. This process yielded elevation and three predictors each for temperature and precipitation factors for downstream analyses.

We used an ensemble SDM approach [44] that included generalized additive models (GAM), generalized boosted models (GBM, i.e., boosted regression trees), and random forest (RF) algorithms [43,82]. We chose these algorithms based upon repeated high performance in modeling trials among available algorithms in biomod2. For GAMs, we used restricted maximum likelihood penalty without interaction via package mgcv [93] to minimize overfitting [79]; 1,000 trees for RF; and default settings for GBM. Throughout, we ran three repetitions each for pseudo-absence permutations, single and ensemble model calibration/validations, and variable importance estimations. We split datasets for training (80%) and testing (20%) per run and evaluated models via relative operating characteristic (ROC) and true skill statistic (TSS) metrics [82,94]. We built ensembles from single algorithm models performing >70% percentile TSS threshold via committee averaging (ca) and proportionally-weighted means (wmean) ensemble algorithms [82]; we again used ROC and TSS to evaluate resulting ensembles.

For future projections, we used the BIOMOD_EnsembleForecasting function in biomod2 to project ensemble models onto new environments per each GCM × SSP × future time scenario. The calculated output was used to visualize distributions of suitable environment in ArcMap as well as predict changes in range size between present and each future scenario. We used Kruskal-Wallis tests to assess model evaluation scores between ensemble algorithms, GCMs, and metrics, respectively. Throughout, we performed all analyses in ArcMap or program R v.4.1.1 [95].

## Results

### Historical shifts

After data refinement, we retained n = 955 *Thamnophis* occurrences for historical analyses (Table 1). Discernable temporal trends across historical records exist for *Thamnophis*. Patterns in central tendency reveal a northwestern progression that has become more directional relative the central highlands—a diverse ecotone between desert ecoregions—and the Mogollon Rim, i.e., the southern extent of the Colorado Plateau (S1 Fig; [64,96]). Mean centers have shifted northwesterly (285.9°) by approximately 5.4 km between pre-1940 to the 1940–1979 periods, and 68.1 km (333.5° NW) to the 1980–present distribution. The area of concentration has contracted across time. Standard ellipse areas reduced by 42.2% between pre-1940 and 1940–1979 periods and another 18.9% from the 1940–1979 period to present. That is, occurrence sampling has become more spatially concentrated.

**Environmental drivers of *Thamnophis* distribution.**   After spatially thinning contemporary occurrences to 1/km$^2$, n = 347 *Thamnophis* records remained available for present-time SDM analyses (Fig 1, Table 1). From those data, we created 20 ensemble models to derive environmental suitability. We detected no differences among ensemble algorithm scores (ca vs wmean) for model sensitivity (H = 3.2, df = 1, $p$ = 0.075), specificity (H = 1.1, df = 1, $p$ = 0.289), and calibration (H = 0.5, df = 1, $p$ = 0.471; S1 Table). Likewise, evaluation metrics (ROC vs TSS) were equivalent in sensitivity (H = 0.0, df = 1, $p$ = 1.000) and specificity (H = 0.1, df = 1, $p$ = 0.910); ROC trended to have higher calibration scores although both metrics scored >0.95 on average (H = 3.7, df = 1, $p$ = 0.053; S1 Table). In summary, all ensemble models yielded satisfactory performance scores.

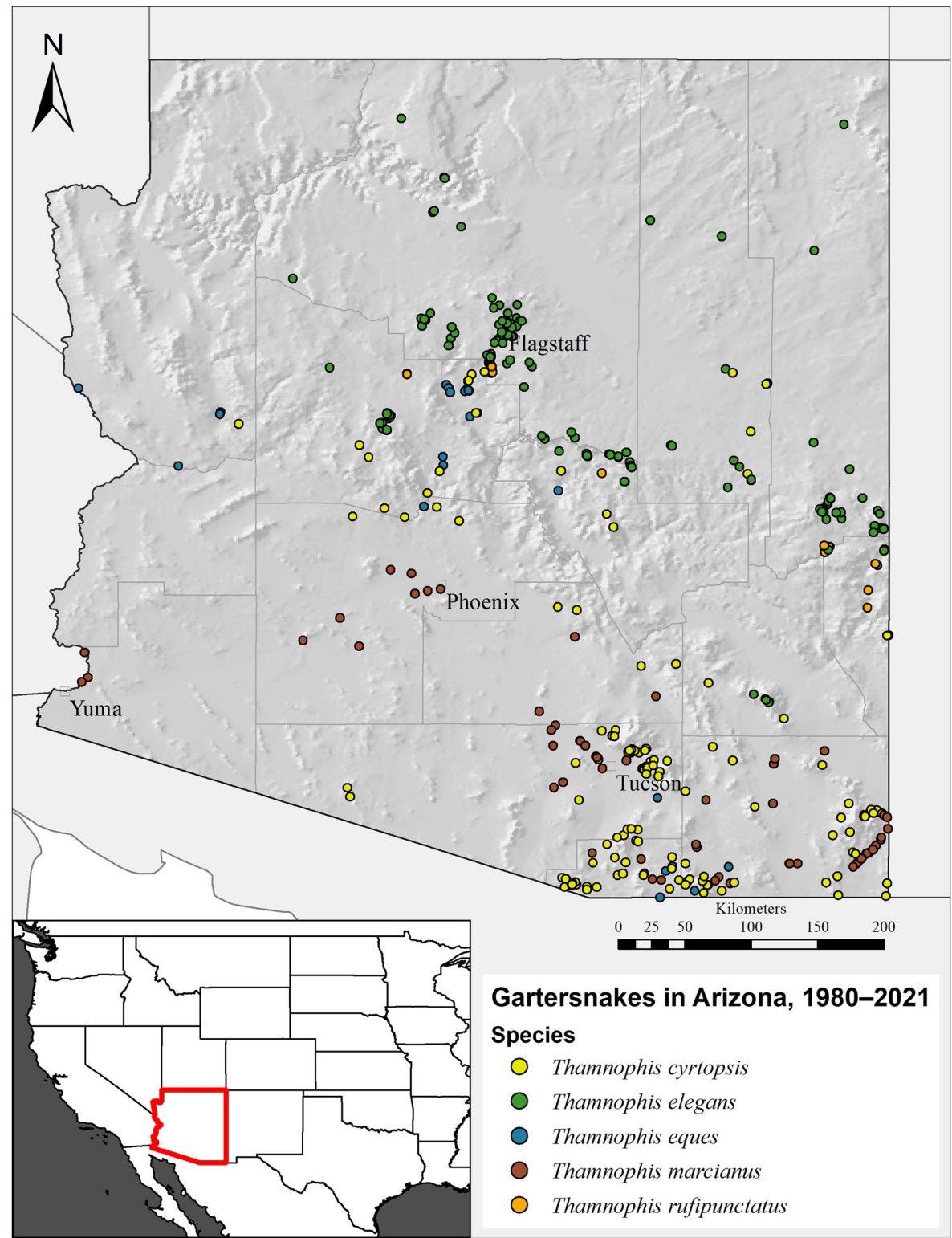

**Fig 1. Distribution of *Thamnophis* gartersnake occurrences in Arizona, 1980–2021.**

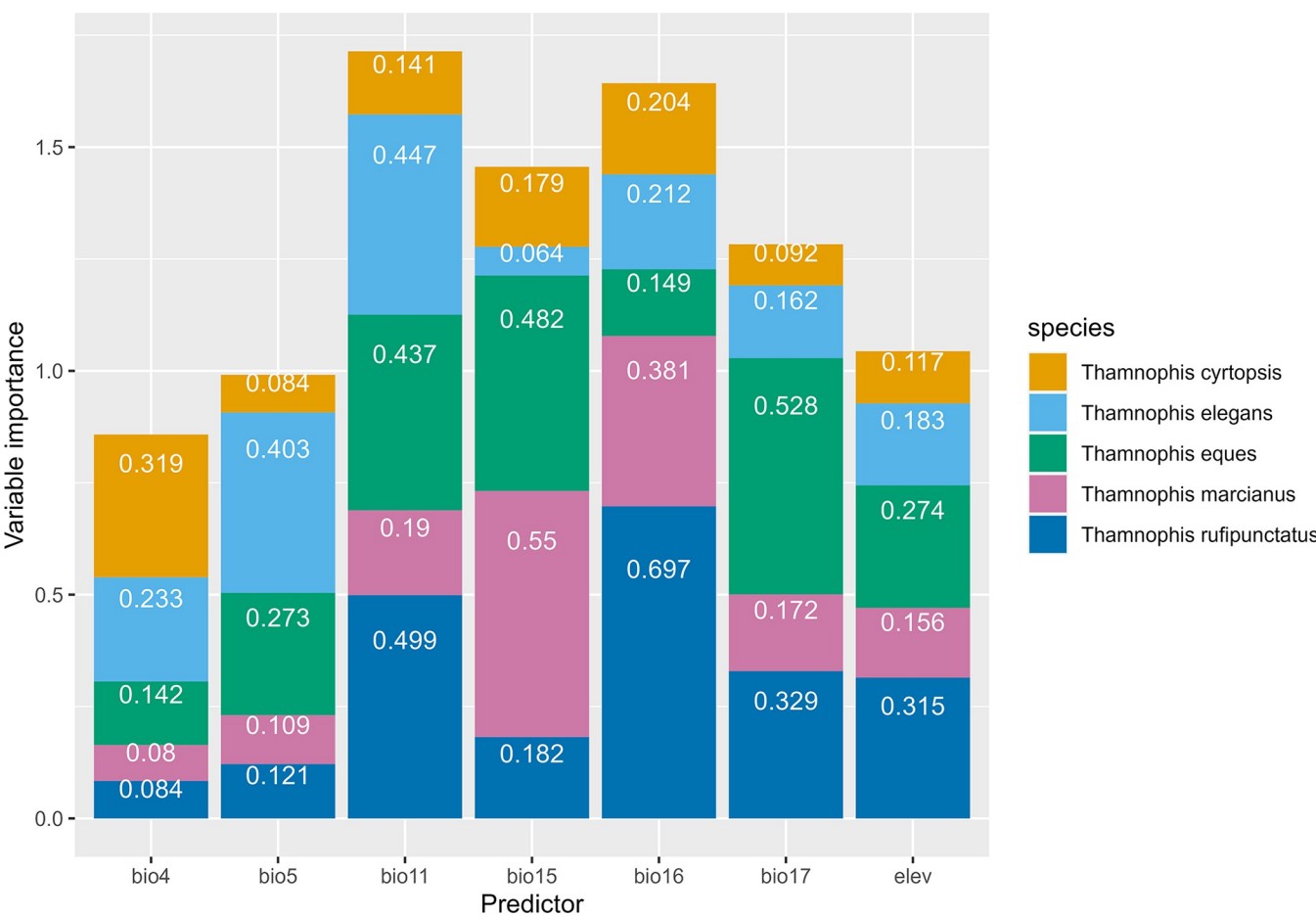

**Fig 2. Predictor importance in species distribution models of environmental suitability for *Thamnophis* gartersnakes in Arizona, 1980–2021.** Variable importance scores range between 0 and 1 per species; column height reflec*t*s cumulative influence across species. Bioclimatic *predictor*s are described in Table 2.

Predictor variable importance was equivalent by species (H = 4.6, df = 4, *p* = 0.334; Fig 2). The most influential predictors overall, however, were mean temperature of coldest quarter (i.e., overwinter dormancy period; Bio11), precipitation during wettest quarter (i.e., summer monsoonal period; Bio16), and precipitation seasonality (Bio15; Table 2). Elevation and maximum temperature of warmest month (Bio5) were the least influential overall, albeit the latter was important to *T. elegans*. However, importance among all predictors were equivalent (H = 5.1, df = 6, *p* = 0.528).

Precipitation variables were most important for *T. eques*, *T. marcianus*, and *T. rufipunctatus* whereas temperature-linked predictors best explained *T. cyrtopsis* and *T. elegans* distributions. Individually, *T. cyrtopsis* were most influenced by temperature seasonality, precipitation of wettest quarter, and precipitation seasonality (Bio4,16,15; mean ± standard deviation (SD) variable importance = 0.162 ±0.082); these related to environmental seasonality and summer precipitation. *Thamnophis elegans* distribution were best predicted by temperature seasonality, max temperature of warmest month, and mean temperature of coldest quarter (Bio4–5,11; mean ±SD variable importance = 0.243 ±0.136). *Thamnophis eques* were best predicted by precipitation of driest quarter, precipitation seasonality, and mean temperature of coldest quarter (Bio17,15,11; mean ±SD variable importance = 0.326 ±0.157); these relate to mean overwinter

**Table 2. Environmental variables used in ensemble species distribution modeling for *Thamnophis* gartersnakes in Arizona, 1980–2021.** *Mean rank* is the mean variable importance score rank (out of 7) among five species, *Thamnophis cyrtopsis*, *T. elegans*, *T. eques*, *T. marcianus*, *T. rufipunctatus*; lower mean rank scores equal greater importance.

| Variable[a] | Description | Mean rank |
|---|---|---|
| Bio4 | temperature seasonality (SD × 100) | 5 |
| Bio5 | max temperature of warmest month | 5.2 |
| Bio11 | mean temperature of coldest quarter | 2.6 |
| Bio15 | precipitation seasonality (coefficient of variation) | 3.6 |
| Bio16 | precipitation of wettest quarter | 3 |
| Bio17 | precipitation of driest quarter | 4 |
| Elev | elevation (meters above sea level) | 4.6 |

[a]Source: WorldClim v. 2 database, bioclimatic variables [74].

temperature and precipitation during the active season. *Thamnophis marcianus* were influenced by precipitation seasonality, precipitation of wettest quarter, and mean temperature of coldest quarter (Bio15–16,11; mean ±SD variable importance = 0.234 ±0.170); these relate to seasonal precipitation, especially during the summer monsoon as well as mean overwinter temperature. *Thamnophis rufipunctatus* were most influenced by precipitation of wettest quarter, mean temperature of coldest quarter, and precipitation of driest quarter (Bio16.11.17; mean ±SD variable importance = 0.318 ±0.219); these relate to mean overwinter temperature and precipitation throughout the active season.

**Future projections of environmental range.** We generated another 240 ensemble models encompassing all future projection scenarios (S2 Fig). Both ensemble algorithms (ca and wmean) were congruent in projected range sizes (e.g., gains, losses, unchanged stable, and unsuitable area) except for present range size (H = 11.7, df = 1, $p < 0.001$); this likely reflects differing per pixel area of unsuitable environment (H = 4.3, df = 1, $p = 0.038$). That is, the majority of the area in the present sampling extent (Arizona) are absent of occurrences and thus considered unsuitable. More importantly, algorithms were not different in predicted gains and losses (all $p > 0.05$). Global climate models were consistent in range projections (all $p > 0.05$) except percent gain (H = 9.0, df = 2, $p = 0.011$), though we note a related trend in per-pixel gain (H = 5.0, df = 2, $p = 0.082$). For SSPs by period scenarios, there were many similarities in range changes ($0.182 \leq p \leq 1.0$) except for percent loss (H = 11.0, df = 3, $p = 0.012$) and percent species range change (H = 7.8, df = 3, $p = 0.049$; S1 Dataset). Overall, GCMs were comparable; HAD exhibited the greatest variation in percent range change per period scenario whereas MRI appeared to be most consistent (S2 Fig). We used MRI output in map projection graphics.

Because wmean and ca algorithms yielded equivalent results but weighted mean ensembles are most often used due to their performance [44], we use wmean hereafter to summarize our findings related to changes in projected environmental suitability (S3 Fig), percent range change (Fig 3), and types of range change by area (Fig 4). All species are predicted to have net reductions in suitable environmental range (mean ± standard error (SE) = -37.1 ±15.5%), though *T. marcianus* is predicted to gain suitable environment (mean = +28.1 ±SE 25.0%; Fig 3). The two species occurring at higher elevations are predicted to lose considerable suitable environment among all SSP scenarios (*T. elegans* mean = -84.5 ±SE 2.8%; *T. rufipunctatus* mean = -84.5 ±SE 7.0%; Fig 3). *Thamnophis eques* is also likely to experience net losses of suitable environment under all SSP scenarios (mean = -34.9 ±SE 16.2%), except into 2090 under SSP126 (mean = +15.0 ±SE 11.5%; Fig 3). Although there was a net negative range change for *T. cyrtopsis* (mean = -10.3 ±SE 26.6%; Fig 3), GCMs for this species exhibited the most

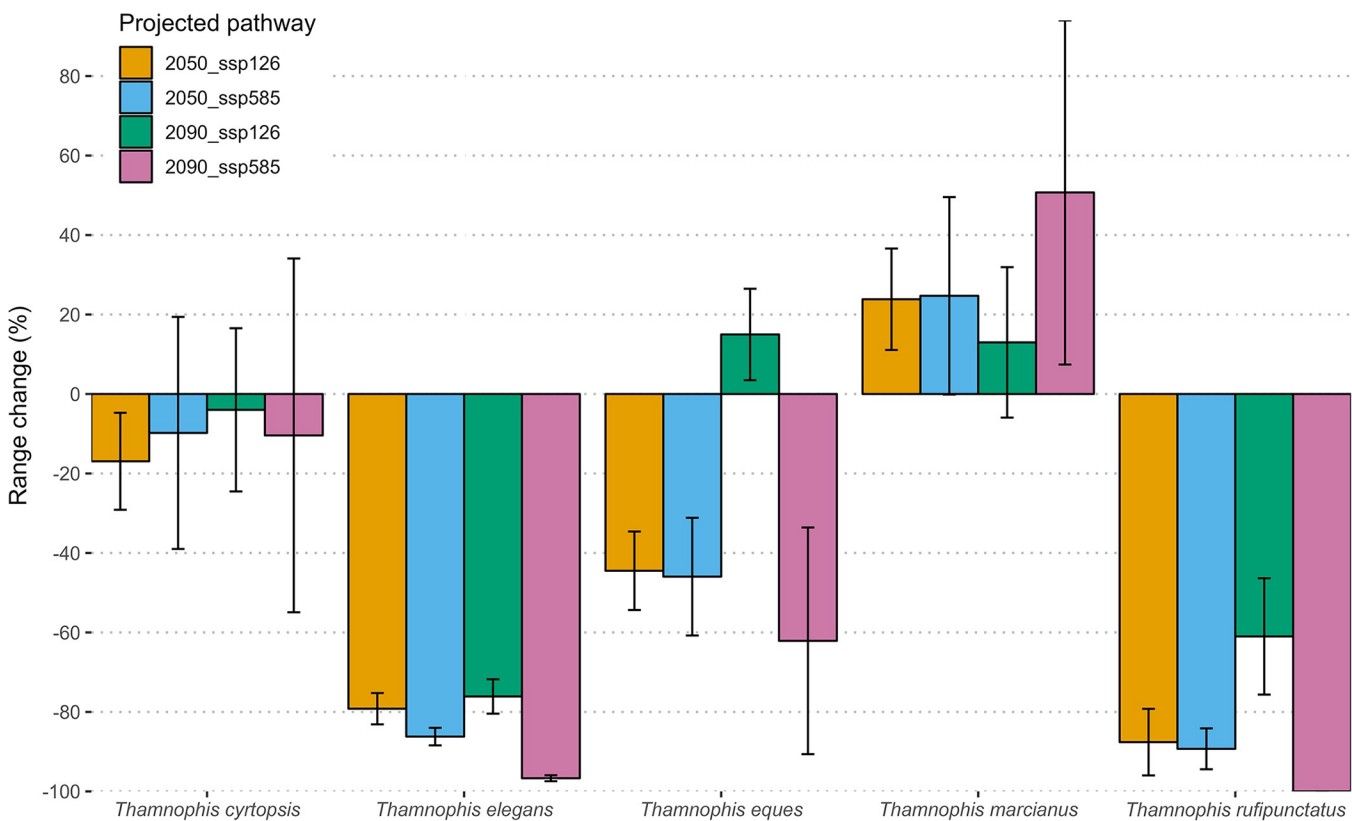

**Fig 3. Estimated future changes in suitable environmental range for *Thamnophis* gartersnakes in Arizona.** Data are derived from the means of weighted-mean ensemble models of three global climate models (GCM), two shared socio-economic pathways (SSP), and two future time period scenarios per species. Bar plots depict mean change plus standard error among GCMs from baseline (0, i.e., present distributional range, 1980–2021). Positive values reflect net increases whereas negative values estimate net reductions.

variation, including estimated positive (HAD) and negative (HAD, MPI, MRI) range changes. Taken together, these projections mostly support our prediction of future declines in suitable environment. Variation exhibited by species and scenario are tabulated in S1 Dataset.

Spatially, favorable and unfavorable area projections were evident under future scenarios, i.e., suitable environment predicted to be lost, retained, or gained (S1 Dataset, Fig 4). The relatively warmer and drier southern part of Arizona appears likely become environmentally unfavorable across time, although this phenomena may be topographically dampened by disjunct, insular mountains—the Madrean Sky Islands [97]. Predicted environmental stability or gains appear linked by elevation, in which lower, montane foothill areas around the Sky Island region reflect losses and areas ascending into higher elevations are more likely to remain stable (Figs 4 and S3). Gains also appear to increase latitudinally, such as around the transition zones of the central highlands and Mogollon Rim region (Figs 4 and S3).

Predicted declines in suitable range were mostly congruent among GCMs for both threatened species. Projections that follow the most optimistic emissions pathway (SSP126) appear less unfavorable to *T. eques* (-14.7 ±SE 10.7%) than under the SSP585 *status quo* pathway (-54.0 ±SE 21.7%; Fig 3, S1 Dataset). More southerly populations may be most affected, with large swaths anticipated to become environmentally unsuitable (Fig 4I–4L). Northern populations of *T. eques*, however, may retain or even gain suitable environment under progressive climate actions (SSP126; Figs 4I–4L and S3K–S3O). *Thamnophis rufipunctatus* likely faces even greater climatic challenges across time. Areas of suitable environment are predicted to become

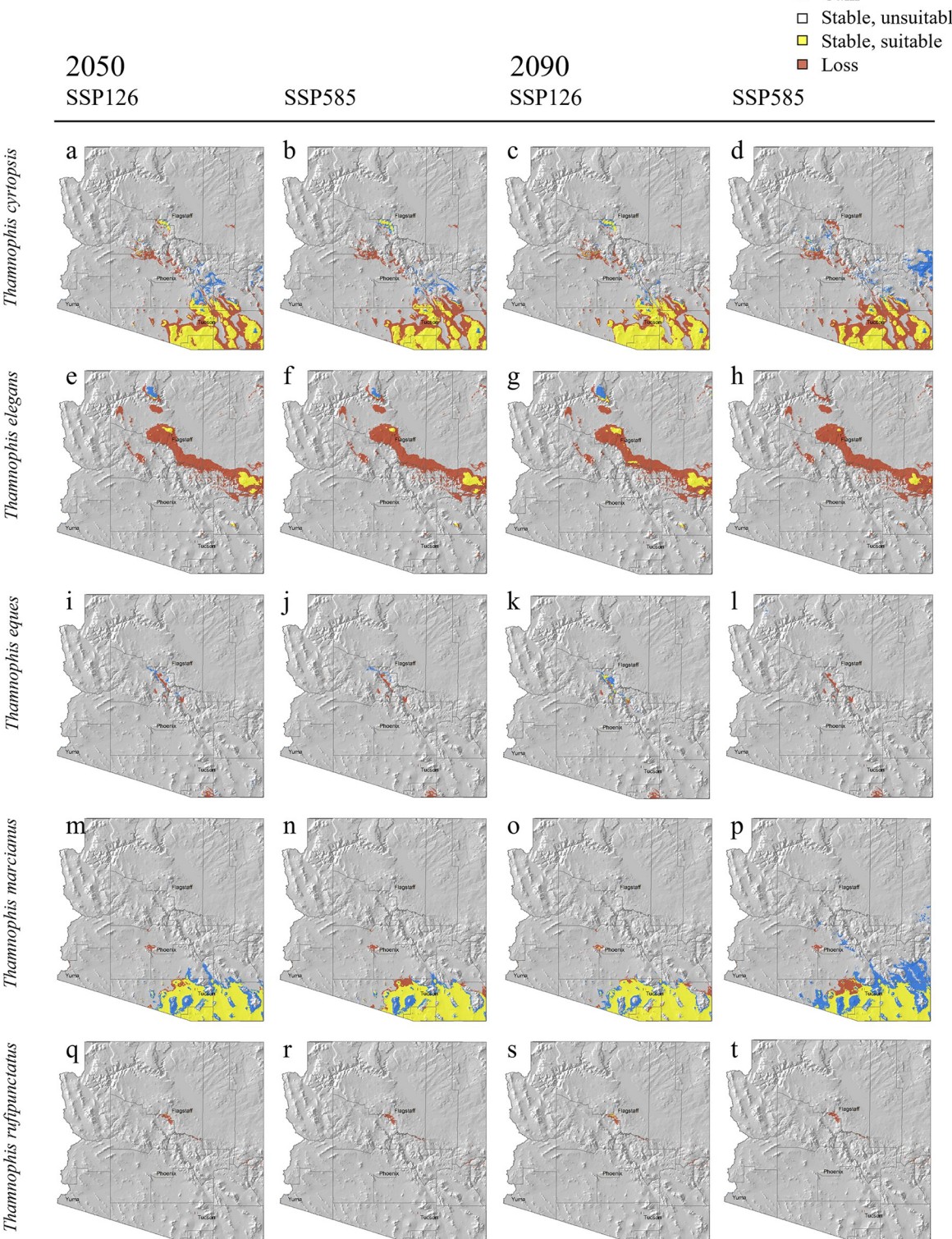

**Fig 4.** a–t. Projected change in range sizes for *Thamnophis* gartersnakes in Arizona. Data are modeled from weighted-means ensemble models generated from the MRI-ESM2-0 global climate model. Data columns represent near future 2050 (i.e., 2041–2060 median) and distant future 2090 (i.e., 2081–2100 median) projections under shared socio-economic pathways (SSPs) that represent most optimistic (SSP126) and pessimistic (SSP585, i.e., *status quo*) emissions-limiting models. Individual map panels for species (a–t) are shown in S4 Fig.

highly reduced and localized (Figs 4Q–4T and S3U–S3Y); only an average of 47.3 pixels (i.e., approx. 47 km$^2$ area) of suitable environment is projected to be gained (S1 Dataset). By the late 21$^{st}$ century, all GCMs predicted a complete loss (-100%) of suitable environment for *T. rufipunctatus* under the nonaction SSP585 pathway, whereas prompt, sustainable emissions reductions may still yield substantial reductions (mean = -74.3 ±SE 11.5%).

## Discussion

Through ensemble species distribution modeling, we provide likelihoods of influential environmental predictors on *Thamnophis* gartersnakes—a model system guild important for energy transfer between aquatic and terrestrial ecosystems. We also projected changes to suitable environmental ranges under various climate change scenarios. Our results show that *Thamnophis* in semiarid regions such as Arizona have shifted northerly across time, various combinations of precipitation and/or temperature factors drive present distributions, and the range of suitable environment are predicted to change—mostly reductions—in the future. Here, we convey a breadth of potential environmental niches available to aridland riparian *Thamnophis* and similar taxa across space and time through the 21$^{st}$ century.

### Modeling considerations and estimating distributional trends

Species distribution models can provide valuable information to researchers, natural resource managers, and decision makers. Careful methodological approaches are important during SDM development stages [98]. For example, we addressed important criteria related to spatial extent considerations [39], ecologically relevant occurrence thinning [83] and pseudo absence parametrization [86,88], predictor selection [39,90,98,99], training and testing data partitioning and calibration [94], and ensemble modeling and evaluation [44,81]. Part of our predictor selection process included using *a priori* information from expertise and relevant literature [92]. Several predictors identified herein, e.g., maximum summer and minimum winter temperature, have worked well for SDMs of snakes in similar latitudes or regions, including congenerics and conspecifics [53,100–104]. Overall, our high performance evaluation scores suggest that our selected variables and ensemble modeling process adequately reflect aridland *Thamnophis*-environmental relationships.

All projected SDMs contain a certain degree of uncertainty and variation in predictions [80,99]. Predicting suitable niche space under future projections may be best achieved by evaluating varying climate-model scenarios, i.e., GCM × SSP × time frames [81,99]. By evaluating multiple scenarios that bound climate change pathway extremes, such as the least and most optimistic SSPs, we reduced the likelihood of uncertainties missed by single scenario models [75,79]. Providing central tendencies and dispersion among a range of outcomes should help guide decision making of conservation practitioners.

Extrapolating SDMs (i.e., transferability) to novel spatial and temporal environments offers additional opportunities to modelers [105,106], albeit caution and deliberation should be exercised when transferring models to new areas [42]. Modeling methods are often focused to original sampling parameters, different species and their traits may require different approaches, and new areas may not spatially or temporally exhibit the same attributes and conditions of the originally assessed space [105,107–109]. We summarize the relative spatial extrapolation potential of our *Thamnophis* SDMs as follows: our models and results are likely to be most transferable where focal taxa have contiguous or proximal neighboring populations (e.g., California, Utah, New Mexico, and Sonora Mexico in this study). The limited distribution of *T. rufipunctatus*, for example, extends into west-central New Mexico, where occupied habitat resembles those in Arizona [110]. We expect that the design and explanatory power of our

 

Arizona-based models would perform comparably among the New Mexican range of *T. rufipunctatus*. Conversely, *T. elegans* is a widely distributed species that ranges into southwestern Canadian latitudes, where habitats and bioclimatic patterns undoubtedly vary from those in Arizona [50,103,111]. Our *T. elegans* models thus may have better transferability to neighboring Utah populations than to those in Alberta. We recommend model flexibility and calibration as geographic × environmental distances increase and thus the potential influence of predictors can also change. Additionally, "stacking" SDMs—overlaying SDMs of multiple taxa to determine concurrent suitable areas [112]—should be considered carefully when differences in distribution and niche occur, such as with *Thamnophis* in Arizona [56]. Rather, best practices to maximize conservation potential may be to focus on syntopic areas of mutually stable suitable environment under current and future projections [113], especially areas with favorable microclimates better buffered against environmental stochasticity [114].

## Projected changes among future scenarios

Much of the semiarid southwestern United States are trending towards hotter and drier conditions [7]. Precipitation events are likely to become more sporadic yet extreme [67,68]. Even under optimistic emission pathways, the southwestern region is likely to face unprecedented drought towards the 22nd century [10]. Primary productivity of functional plant groups will likely be affected [9,69,107], with subsequent cascading effects across trophic dynamics [8,33]. In aridland systems, much biodiversity is concentrated around perennially or seasonally available surface water, such as riparian zones [115,116]. Anthropogenic expansion and water resource demands are predicted to rise across time [5], which will further exacerbate water availability [117,118]. Alteration or depletion of aquatic networks can drastically and negatively change riparian habitat and stress biota that rely upon it [16,106,119], including *Thamnophis* [101].

Our findings that *Thamnophis* in Arizona appeared to shift northerly in latitude in recent history lends support to our first hypothesis. We found that contemporary distributions of *Thamnophis* are best explained by overwinter temperature, summer precipitation, and precipitation seasonality. Because the species in our study area have varying distributions, niches, and status [56], we break down prospective summaries for each taxon under future climate scenarios. We find it interesting that *T. maricianus* is predicted to gain suitable environment, whereas all other congenerics face reduced suitability. In our study area, *T. marcianus* is the only species that is common, a dietary generalist, and predominantly occupies various riparian wetland and riverine systems at lower elevations (e.g., desertscrub and arid grasslands) [120]. The other four species are either rare or usually associated with habitats above desert valley floors. Our *T. maricianus* models identified a lesser degree of sensitivity to both dry season precipitation (e.g., late spring to early summer prior to monsoonal rains) and overwinter temperature. Overwinter temperature was strongly associated with taxa facing the greatest predicted losses of suitable environment, especially at higher elevations. Precipitation seasonality was strongly influential for *T. marcianus*, suggesting plasticity for temporal variability in rainfall. Although we did not assess intrinsic traits, our findings corroborate the adaptive variation in this species [121,122]. Given their adaptive potential, current population trends [56], and future projections (here), *T. marcianus* may be more flexible to changing conditions than its aridland congenerics.

Precipitation during the driest season was among the most important factors for both threatened species (*T. eques* and *T. rufipunctatus*) but not for other congenerics. Overwinter temperature was also notably important. Seasonal relationships are not surprising given that *T. eques* is known to shift behaviors between active and inactive seasons [123], and often occupies aquatic drainages with flow regimes regulated by precipitation pulses [124]; and some *T.*

 

*rufipunctatus* life history phenology (e.g., parturition timing) are linked to monsoonal precipitation and subsequent changes to aquatic habitat [50,125]. Other SDM studies for *T. eques* projected greater sensitivity to environmental extremes and, in support of our findings, that overwinter temperature is an important predictor [53,101]. Giermakowski and others [101] also found summer precipitation to be important for both threatened gartersnakes and, in addition, maximum summer temperature for *T. eques*. Our ensemble modeling differs from that study in that we found temperature to be more important in winter than summer for both rare gartersnakes. All studies, however, estimated substantial declines in suitable areas, 7–62% for *T. eques* and 32–62% reductions for *T. rufipunctatus* by the end of the century. Further fluctuation in the timing, intensity, and duration of seasonal environmental patterns are likely to adversely affect populations of these snakes.

Climatic shifts towards warmer summers and winters may negatively affect *T. elegans* the most. The top explanatory variables for this high-elevation generalist were all temperature related. Substantial projected reductions in suitable environment from our SDMs are comparable to congeneric *T. scalaris*, another common, high-elevation species in Mexico [53]. More broadly, the predictions for *T. elegans* align with findings occurring at a global scale in which range contractions and local extinctions often arise at the "warm edge" of a species range [58]; Arizona largely represents the warmer, southern extent of the widely distributed *T. elegans* [50,110,111].

*Thamnophis cyrtopsis*—a dietary generalist often occupying sloped, rocky riparian habitats [50,57,126,127]—presented the most variability in projected range change, resulting in either estimated expansion or contraction pending the future scenario. Seasonality of predictors most influenced distribution for this species (Fig 2, Table 2). These results somewhat contrast to populations along central Mexico's Trans-Mexican Volcanic Belt, where *T. cyrtopsis* was found to be more sensitive to the extremes, including minimum winter temperature and warmest summer temperature [53]. All, however, predict a decline in suitability with increasing temperature. Here, projected trends for *T. cyrtopsis* appear topographically linked with suitability favoring an upward ascent (see S3 Fig). This suggests that habitats in ascending altitudes (e.g., Madrean evergreen woodlands) may offer the most optimal refugia in the future, especially areas more buffered against extreme climatic shifts, and that lower foothill populations likely face greater risks. Given a sensitivity to seasonal environmental relationships [this study] and extreme disturbances [57], selective forces may pressure this taxon to ascend upward in elevation similar to other montane taxa [59,60,128–130]. There is only so far a taxon can advance, however, especially when reliant on riparian networks and available surface water [57].

## Future directions

In semiarid regions such as the southwestern United States, significant shifts in upward movement and phenology of flora have already occurred [129,130]. For representative species that link terrestrial and aquatic habitats like *Thamnophis*, climatic stressors and challenges are likely to add pressure for adaptation. Common generalists like *T. cyrtopsis* and *T. marcianus* may be better able to adjust to shifting and reestablished aquatic regimes (e.g., effluent-recharged Santa Cruz River; [131]) or disperse to more favorable environments in the long run. In the short term, however, extreme disturbances and rapid environmental changes can still yield swift, adverse consequences even for generalist taxa [57,127]. As climates and environments change, surveillance of demographic trends and responses to stimuli by common species are worth monitoring by natural resource managers tasked with anticipating and mitigating threats to biodiversity [5,54,132,133].

Modeling for rare taxa is understandably challenging. Species with narrow ranges may face greater risks if changes result in fragmentation or elimination of favorable environment

[79,84,106,134]. High-elevation, cold-tolerant *Thamnophis* are more vulnerable to rising temperature [53]. Increased temperature is likely to adversely affect *T. rufipunctatus*—a specialist endemic to cold, clear-water riparian zones in the Gila River Watershed in Arizona and New Mexico [110]. Both threatened *T. eques* and *T. rufipunctatus* have had considerable population reductions since the late 20[th] century, with declines attributed to habitat loss or degradation, effects from invasive species, wildfire and other disturbances [62,110,124,135]. Our projections of substantial reduction in suitable environment across all future scenarios are concerning and may compound stressors already faced by populations. Further detriment to small populations heightens the risks related to demographic stochasticity—including but not limited to effects from genetic drift, inbreeding, and reduced heterozygosity [53,54,63].

A conservation-driven application of SDMs is to identify areas predicted to remain or become "climate refugia" [136], i.e., suitable safe places across time. Locating high-likelihood areas of environmental suitability under future change is vital for sensitive taxa with limited distribution. Existing management areas with umbrella-species coverage [137] or Critical Habitat designations, for example, can be compared for overlap with SDM output. Our findings suggest that for *T. eques*, refugia may persist around their northerly, central-Arizona populations [124,138] whereas populations near the U.S. border with Mexico may face more disadvantageous climatic challenges. This could further restrict connectivity potential with populations in Mexico [53,63]. For *T. rufipunctatus*, areas projected to sustain suitable environment are likely to become highly confined, and the most optimal areas may be in central Arizona (e.g., Verde River Watershed) and extreme eastern Arizona (e.g., Upper Gila River Watershed; [110,139]). Managing connectivity of the latter with populations in western New Mexico (see [63]) may be imperative to mitigate heightened extinction risks for a taxon that has little to nowhere to go. Applying other conservation actions, such as genetic rescue via translocations of *ex situ* or *in situ* stock [140] could help stave off existing burdens of already threatened species [125,134,141], including threatened *T. eques* and *T. rufipunctatus* [63].

Proactive and best practice conservation initiatives might be to rigorously assess population demography, identify and manage priority habitat areas—especially where favorable under future projections, and implement practical, longitudinal conservation actions. Our modeling outcomes highlight potential reductions in suitability under future climate scenarios but also the importance of conserving connectivity that mitigates further detriment to biodiversity. Additional data and SDM refinement presents opportunities to further examine spatiotemporal uncertainties in distribution and projections [142], especially as habitats and land-uses change, which is likely to benefit our understanding of species-environmental relationships under anthropogenic climate change.

## Conclusion

*Thamnophis* are an important intermediate predator guild linking terrestrial and freshwater networks. Gartersnakes serve as environmentally sensitive bioindicators and may reflect the effects of climatic stressors induced on other riparian biota with similar life histories. Our rigorous varying-scenario SDMs identified important environmental predictors that drive present distributions of gartersnakes among Arizona's diverse ecosystems and their favorable or unfavorable outlook under future climate change. Our hypothesis of predicted declines in suitable environment are supported for three species, of mixed outcomes for one species, and seemingly rejected for *T. marcianus* that has a more favorable outlook. The information herein can help guide proactive conservation strategies now and into the future. Consideration of evidence-based conservation approaches are paramount to stave off extinction risks as the climate warms and dries.

## Supporting information

**S1 Fig. Directional distribution trends of *Thamnophis* gartersnake occurrences in Arizona.** Ellipses represent the central tendencies (mean center ±1 standard deviation in each XY coordinate) of the data. Three time periods are categorized to reflect historical change.
(PDF)

**S2 Fig. Estimated range changes across time for five *Thamnophis* gartersnake species in Arizona under three global climate models, two 20-year periods (median years 2050, 2090), and two shared socio-economic pathways (SSP).** Data are derived from weighted-mean algorithms of ensemble species distribution models. Bars represent means ± standard error. Pathways include the most optimistic (SSP126) and pessimistic (SSP585) emissions-limiting models.
(PDF)

**S3 Fig. a–y. Environmental suitability maps for Thamnophis gartersnakes in Arizona.** Projections based on weighted-means ensemble models generated from the MRI-ESM2-0 global climate model. Data columns represent present (1980–2021), near future ("2050", i.e., 2041–2060 median), and distant future ("2090", i.e., 2081–2100 median) projections of two future shared socio-economic pathways (SSP) that represent optimistic (SSP126) and pessimistic (SSP585, i.e., status quo) emissions-limiting models.
(PDF)

**S4 Fig. . Individual panels (a–t) from 'Fig 4 Projected change in range sizes for Thamnophis gartersnakes in Arizona'.** Data are modeled from weighted-means ensemble models generated from the MRI-ESM2-0 global climate model. Changes based on present time modeling (1980–2021). Color values: yellow = suitable, stable; blue = suitable gained; red = lost, unsuitable; gray = unsuitable.
(PDF)

**S1 Table. Ensemble species distribution models for *Thamnophis* gartersnakes in Arizona, 1980–2021.** Ensemble models are comprised of multiple single-algorithm models (GAM, GBM, and RF) with true skill statistics (TSS) scores >70%. Ensemble algorithms (*EM.algo*) include committee averaging (*Ca*) and weighted mean (*wmean*; [82]), evaluation metrics (*Eval.met*) include relative operating characteristic (*ROC*) and *TSS*, and reported values include ensemble model sensitivity (*sens*, i.e., true positive rate), specificity (*spec*, i.e., true negative rate), and calibration (*cal*).
(PDF)

**S1 Dataset. Datasets used in species distribution modeling of *Thamnophis* gartersnakes in Arizona.** Sheets included represent 1) metadata descriptions; 2) refined occurrence records, excluding rare species; 3) ensemble SDM evaluation scores and environmental variable importance scores per species; and 4) suitable environment range change output from ensemble species distribution modeling per species.
(XLSX)

## Acknowledgments

We thank M. Bogan, M. Culver, I.M. Vela-Vargas, S. Wells, and C. Wissler for valuable feedback on an earlier version of this manuscript. We are grateful to M. Guéguen and R. Patin for helpful assistance with biomod2 code formatting. We acknowledge the World Climate

Research Programme, which, through its Working Group on Coupled Modelling, coordinated and promoted CMIP6.

## Author Contributions

**Conceptualization:** Brian R. Blais, John L. Koprowski.

**Data curation:** Brian R. Blais.

**Formal analysis:** Brian R. Blais.

**Investigation:** Brian R. Blais.

**Methodology:** Brian R. Blais.

**Project administration:** Brian R. Blais, John L. Koprowski.

**Resources:** Brian R. Blais, John L. Koprowski.

**Software:** Brian R. Blais.

**Supervision:** John L. Koprowski.

**Validation:** Brian R. Blais.

**Visualization:** Brian R. Blais.

**Writing – original draft:** Brian R. Blais.

**Writing – review & editing:** John L. Koprowski.

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
