## [Decision Letter · Decision Letter 0]

23 Jan 2024

PONE-D-23-43716Modeling a hot, dry future: substantial range reductions in suitable environment projected under climate change for a semiarid riparian predator guildPLOS ONE

Dear Dr. Blais,

Thank you for submitting your manuscript to PLOS ONE. After careful consideration, we feel that it has merit but does not fully meet PLOS ONE’s publication criteria as it currently stands. Therefore, we invite you to submit a revised version of the manuscript that addresses the points raised during the review process.

We look forward to receiving your revised manuscript.

Kind regards,

Randeep Singh

Academic Editor

PLOS ONE

Journal Requirements:

3. To comply with PLOS ONE submissions requirements, please provide the following information in the Methods section of the manuscript and in the “Ethics Statement” field of the submission form (via “Edit Submission”): 

*  Please indicate whether an animal research ethics committee prospectively approved this research or granted a formal waiver of ethics approval.*  Please enter the name of your Institutional Animal Care and Use Committee (IACUC) or other relevant ethics board. Also include an approval number if one was obtained.

*   If anesthesia, euthanasia, or any kind of animal sacrifice is part of the study, please include briefly in your statement which substances and/or methods were applied.

For additional information about PLOS ONE submissions requirements for ethics oversight of animal work, please refer to http://journals.plos.org/plosone/s/submission-guidelines#loc-animal-research 

4. In this instance it seems there may be acceptable restrictions in place that prevent the public sharing of your minimal data. However, in line with our goal of ensuring long-term data availability to all interested researchers, PLOS’ Data Policy states that authors cannot be the sole named individuals responsible for ensuring data access (http://journals.plos.org/plosone/s/data-availability#loc-acceptable-data-sharing-methods).

Reviewers' comments:

Reviewer's Responses to Questions

**Comments to the Author**

1. Is the manuscript technically sound, and do the data support the conclusions?

Reviewer #1: Partly

Reviewer #2: Yes

2. Has the statistical analysis been performed appropriately and rigorously? 

Reviewer #1: Yes

Reviewer #2: Yes

3. Have the authors made all data underlying the findings in their manuscript fully available?

Reviewer #1: Yes

Reviewer #2: Yes

4. Is the manuscript presented in an intelligible fashion and written in standard English?

Reviewer #1: No

Reviewer #2: Yes

5. Review Comments to the Author

Reviewer #1: Dear Editor

The manuscript about new records should be including comprehensive data on distribution patterns, ecology and taxonomy of target taxon. Moreover, biodiversity approach provide motivation for readers as well conservation managers. Accordingly, the manuscript should be improved to qualify a scientific manuscript including:

General comments

Scientific names should be written based on taxonomical guidelines (Genus, species and authors)

The table, images and graphs can be complete the puzzle of study

Introduction. The author(s) should be improve it including:

Comprehensive literature review

The hypothesis, justification, necessity and aims of the study

Material and Methods should be classified to

Study area: Geography, geomorohology, geology, climate and pedology of area.

Discussion

Discussion is weak. Accordingly must be improved significantly.

Comparison of results by last studies

Detailed analysis

The main achievements must be described by conservation and biodiversity approach

What are the main and key results of study?

Best Regards

Reviewer #2: The authors have predicted the changes in the distribution and range of suitable habitats for five species of gartersnakes by the species occurrence data during the past periods until present period and the environmental variables and using several predictive models. Data provided are valuable. However, it can be written in a better way and more concentrated, which I have specified in the attached file.

6. PLOS authors have the option to publish the peer review history of their article (what does this mean?). If published, this will include your full peer review and any attached files.

Reviewer #1: No

Reviewer #2: No

---

## [Author Response · Author response to Decision Letter 0]

19 Feb 2024

Please see attached 'response to reviewers' document

---

## [Decision Letter · Decision Letter 1]

16 Apr 2024

Modeling a hot, dry future: substantial range reductions in suitable environment projected under climate change for a semiarid riparian predator guild

PONE-D-23-43716R1

Dear Dr. Blais,

We’re pleased to inform you that your manuscript has been judged scientifically suitable for publication and will be formally accepted for publication once it meets all outstanding technical requirements.

Kind regards,

Daniel de Paiva Silva, Ph.D.

Academic Editor

PLOS ONE

Additional Editor Comments (optional):

I am pleased to accept your manuscript for publication in PLoS One.

Congratulations!

Best regards,

Daniel Silva

Reviewers' comments:

Reviewer's Responses to Questions

**Comments to the Author**

1. If the authors have adequately addressed your comments raised in a previous round of review and you feel that this manuscript is now acceptable for publication, you may indicate that here to bypass the “Comments to the Author” section, enter your conflict of interest statement in the “Confidential to Editor” section, and submit your "Accept" recommendation.

Reviewer #1: (No Response)

Reviewer #2: (No Response)

2. Is the manuscript technically sound, and do the data support the conclusions?

Reviewer #1: Yes

Reviewer #2: (No Response)

3. Has the statistical analysis been performed appropriately and rigorously? 

Reviewer #1: Yes

Reviewer #2: (No Response)

4. Have the authors made all data underlying the findings in their manuscript fully available?

Reviewer #1: Yes

Reviewer #2: (No Response)

5. Is the manuscript presented in an intelligible fashion and written in standard English?

Reviewer #1: Yes

Reviewer #2: (No Response)

6. Review Comments to the Author

Reviewer #1: Dear Editor

The manuscript including valuable data for readers as well conservation managers. The comments have been done on, so it is acceptable

Best Regards

Reviewer #2: (No Response)

7. PLOS authors have the option to publish the peer review history of their article (what does this mean?). If published, this will include your full peer review and any attached files.

Reviewer #1: **Yes: **Ahmad Reza Mehrabian

Reviewer #2: No
